# Prenatal Bisphenol a Exposure and Postnatal Trans Fat Diet Alter Small Intestinal Morphology and Its Global DNA Methylation in Male Sprague-Dawley Rats, Leading to Obesity Development

**DOI:** 10.3390/nu14122382

**Published:** 2022-06-08

**Authors:** Sarah Zulkifli, Noor Shafina Mohd Nor, Siti Hamimah Sheikh Abdul Kadir, Norashikin Mohd Ranai, Noor Kaslina Mohd Kornain, Wan Nor I’zzah Wan Mohd Zain, Mardiana Abdul Aziz

**Affiliations:** 1Institute of Medical Molecular Biotechnology, Faculty of Medicine, Universiti Teknologi MARA (UiTM), Cawangan Selangor, Kampus Sungai Buloh, Jalan Hospital, Sungai Buloh 47000, Selangor, Malaysia; sarahzulkifli0913@gmail.com; 2Department of Paediatrics, Faculty of Medicine, Universiti Teknologi MARA (UiTM), Cawangan Selangor, Kampus Sungai Buloh, Jalan Hospital, Sungai Buloh 47000, Selangor, Malaysia; shikin_ranai@uitm.edu.my; 3Institute for Pathology, Laboratory and Forensic Medicine (I-PPerForM), Faculty of Medicine, Universiti Teknologi MARA (UiTM), Cawangan Selangor, Kampus Sungai Buloh, Jalan Hospital, Sungai Buloh 47000, Selangor, Malaysia; sitih587@uitm.edu.my; 4Department of Biochemistry and Molecular Medicine, Faculty of Medicine, Universiti Teknologi MARA (UiTM), Cawangan Selangor, Kampus Sungai Buloh, Jalan Hospital, Sungai Buloh 47000, Selangor, Malaysia; wnizzah@uitm.edu.my; 5Department of Pathology, Faculty of Medicine, Universiti Teknologi MARA (UiTM), Cawangan Selangor, Kampus Sungai Buloh, Jalan Hospital, Sungai Buloh 47000, Selangor, Malaysia; noorkaslina@uitm.edu.my (N.K.M.K.); mardiana2507@uitm.edu.my (M.A.A.)

**Keywords:** bisphenol A, prenatal exposure, postnatal trans fat diet, small intestine, morphology, global methylation, obesity

## Abstract

In this study, we aimed to determine whether a postnatal trans fat diet (TFD) could aggravate prenatal bisphenol A (BPA) exposure effects on offspring’s small intestine and adulthood obesity, due to the relatively sparse findings on how the interaction between these two variables interrupt the small intestinal cells. Twelve pregnant rats were administered with either unspiked drinking water (control; CTL) or BPA-spiked drinking water throughout pregnancy. Twelve weaned pups from each pregnancy group were then given either a normal diet (ND) or TFD from postnatal week (PNW) 3 until PNW14, divided into control offspring on normal diet (CTL-ND), BPA-exposed offspring on normal diet (BPA-ND), control offspring on trans fat diet (CTL-TFD), and BPA offspring on trans fat diet (BPA-TFD) groups. Body weight (BW), waist circumference, and food and water intake were measured weekly in offspring. At PNW14, small intestines were collected for global DNA methylation and histological analyses. Marked differences in BW were observed starting at PNW9 in BPA-TFD (389.5 ± 10.0 g; *p* < 0.05) relative to CTL-ND (339.0 ± 7.2 g), which persisted until PNW13 (505.0 ± 15.6 g). In contrast, water and food intake between offspring were significantly different (*p* < 0.01–0.05) at earlier ages only (PNW4–6 and PNW7–9, respectively). Furthermore, substantial differences in the general parameters of the intestinal structures were exclusive to ileum crypt length alone, whereby both BPA-ND (150.5 ± 5.1 μm; *p* < 0.001), and BPA-TFD (130.3 ± 9.9 μm; *p* < 0.05) were significantly longer than CTL-ND (96.8 ± 8.9 μm). Moreover, BPA-ND (2.898 ± 0.147%; *p* < 0.05) demonstrated global small intestinal hypermethylation when compared to CTL-ND and CTL-TFD (1.973 ± 0.232% and 1.913 ± 0.256%, respectively). Prenatal BPA exposure may significantly affect offspring’s physiological parameters and intestinal function. Additionally, our data suggest that there might be compensatory responses to postnatal TFD in the combined BPA prenatal group (BPA-TFD).

## 1. Introduction

Bisphenol A (BPA), a well-known endocrine disruptor, has always been used as the base material for epoxy resins and polycarbonate plastic productions [1,2,3]. Various concentrations of BPA have been detected in foetal cord blood, liver, urine, amniotic fluid, and placenta [4,5]. These concentrations indicate that foetuses are most likely exposed to BPA through maternal uptake [6,7,8], and the main concern of BPA should be on foetuses since they have a lower hepatic glucuronidation rate compared to healthy adults [9]. Moreover, growing evidence has pointed out the role of BPA in the Developmental Origins of Health and Diseases hypothesis, in which foetal organ development is hugely influenced by maternal diet and lifestyle [10,11,12]. It has been suggested that unfavourable intrauterine environments including environmental exposures and malnutrition may change the function and structure of organs via the alterations of the foetal epigenome [13,14]. These molecular changes could continue lifelong, contributing to the development of adulthood metabolic diseases, including obesity. Notably, a recent study reported the epidemiology of obesity up to 2015 using the data from the Global Burden of Disease Study, which estimated a worrying number of overweight and obese adults over 20 years, which were 1.9 billion and 609 million, respectively [15], surpassing the predicted numbers for 2030 reported in 2008 [16].

A ‘weight gain promoting’ environment—for example, the presence of BPA, which has been considered an obesogen—could amplify the effects of the cumulative contribution of a large number of obesity-related genes, which act via epigenetic changes [17,18]. Three models, human mother–child pairings from the LINA cohort, in vivo mice, and in vitro adipocytes, have been used to study the effect of BPA exposure during the gestational period on body weight development in the long term [19]. A mediator analysis revealed that high prenatal BPA levels were associated with MEST (mesoderm-specific transcript) promoter hypomethylation, upregulation of MEST RNA, and higher BMI z-scores of 1-year-old infants. Similar findings were observed in mice prenatally exposed to 5 μg/mL BPA, in addition to a marked increase in fat mass at ten weeks. Respective decreased and increased MEST promoter methylation and expression, and enhanced adipogenesis (depicted by more lipid droplets formed and upregulation of adipocyte-relevant genes, e.g., PPARγ, LPL, SREBF1, IRS2 and FASN) were also observed in differentiated human mesenchymal stem cells exposed to 10 and 50 μM BPA. Hypomethylation of fggy promoter due to BPA exposure resulted in Fggy expression, whereby the expression strength was positively correlated with the body weight and gonadal fat mass of male mice [20]. Altogether, these findings suggest that perturbed DNA methylation may be one of the underlying causes of adult-onset obesity following BPA exposure in utero.

Adaptive changes in the intestine to the amount of luminal nutrients play a substantial role in obesity development [21]. It has been shown that cell number, crypt depth, intestinal length, absorption, and permeability were increased in obese humans and animals. Moreover, BPA has been shown to negatively influence the intestinal barrier function [22,23]. In one study, prolonged exposure to 50 μg/kg BPA daily (from childhood to adulthood) resulted in intestinal hyperpermeability in mice (Feng et al., 2019). Meanwhile, higher BPA concentration (5000 μg/kg bw/day) reduced the basal intestinal permeability in female offspring exposed to the compound perinatally [24]. In addition, prolonged exposure of tolerable daily intake (50 μg/kg/day) of BPA in mice destroyed the colonic epithelium morphology—severe cellular damage, indentations, and extensive gland separation—as well as elevated the pathology score [25]. Increases in colonic mucosa and plasma endotoxin, D-lactate, diamine peroxidase, and zonulin levels, which indicated intestinal permeability, were also observed. Furthermore, BPA exposure in developing pigs caused changes to the neuronal marker levels of the intrahepatic parasympathetic nerves, which are related to obesity, diabetes, and behavioural and psychological illnesses in children [26].

Importantly, previous studies have demonstrated that the postnatal environment could amplify the final phenotypic outcomes programmed by early developmental insults [27,28,29]. Nonetheless, to the best of our knowledge, no studies have reported on the responsiveness of intestinal function to postnatal insults, ensuing the metabolic disruptions programmed by prenatal BPA exposure. Moreover, higher consumption of trans fat, which is frequently used in processed food to increase its shelf life, has been associated with obesity and its co-morbidities, as well as intestinal inflammation [30,31,32,33]. Hence, to test the hypothesis whether postnatal trans fat diet (TFD) could exacerbate prenatal BPA exposure impacts on the structure and function of offspring’s small intestine, pregnant rats were treated with BPA-spiked drinking water until delivery. Weaned offspring were then fed with either a normal diet or TFD until adulthood, whereby physiological parameters for obesity were monitored weekly from weaning to adulthood. Meanwhile, offspring’s intestinal morphology and global DNA methylation were assessed at adulthood.

## 2. Materials and Methods

### 2.1. Chemicals

In this study, 0.05 g of BPA (Sigma-Aldrich, Catalogue no. 239658-50G, Darmstadt, Germany) was dissolved together with Tween-80 (Sigma-Aldrich, Catalogue No. 8.22187.1000, Darmstadt, Germany), which made up 0.25% of the final volume (1 L) of BPA water. The presence of vehicle Tween-80 facilitates the absorption of BPA in the gut, as it has poor solubility. The concentration of the final working solution was 5 mg/kg/day, and BPA-spiked drinking water was given to pregnant rats only, for 20 to 22 consecutive days.

### 2.2. Experimental Protocol

This study was conducted according to the guidelines of the Animal Research and Ethics Committee of Universiti Teknologi MARA (UiTM) (approval number UiTM CARE: 294/2020 (7 February 2020)). The timeline of the experiment from early life to adulthood is illustrated in Figure 1. Pregnant Sprague-Dawley rats were arbitrarily assigned to either control (CTL) or 5 mg/kg/day BPA (BPA) groups (*n* = 6/treatment). We chose 5 mg/kg/day BPA (a dose ten times lower than the Lowest Observed Adverse Effect Level of BPA, as established by the US Environmental Protection Agency) because several studies have demonstrated the deleterious effects of perinatal BPA in multiple organs at this dosage [24,34,35,36]. Pregnant females were orally administered with BPA via glass-bottled drinking water throughout pregnancy, as previously described [37]. Pups were delivered normally and were weaned off at postnatal week (PNW) 3. Offspring were then fed with either a normal diet (ND; Gold Coin, Klang, Malaysia) or TFD (Research Diets, New Brunswick, NJ, USA) up to PNW14 (Appendix A). The TFD contained 25 kcal% fat, and the main trans fatty acid isomer was trans-18:1, n-6 (elaidic acid). Pregnant rats and fully weaned offspring were housed individually under standard conditions (22 °C, 12 h light/dark cycle). All animals were fed ad libitum throughout the experiment.

### 2.3. Physiological Parameters for Obesity

BW, waist circumference (WC), water intake, and food intake of offspring were measured at weekly intervals from postnatal week PNW3 to PNW13. Readings for body weight and waist circumference were taken on two consecutive days and were then averaged. Body weight was measured by using a weighing balance, and waist circumference was measured by using measuring tape (skin at the back of the neck was held taut during the latter procedure so that the rat remained stationary when measurement was taken). Meanwhile, food and water intake were measured by taking the readings for both parameters on two consecutive days (in 24 h time). The difference between the readings of the day before and the day after was then calculated. Rats were placed in metabolic cages for the food and water intake measurements to eliminate bias in the readings due to food/water spillage. A weighing balance and measuring cylinder were used to measure the food and water intake, respectively.

### 2.4. Organ Tissue Collection

Adult offspring were euthanised by cardiac puncture on PNW14. Intestinal samples were surgically removed and rinsed with phosphate-buffered saline. Samples were either snap-frozen in liquid nitrogen and stored at −80 °C for global DNA methylation analysis, or kept in 10% neutral buffered formalin for morphological study.

### 2.5. Histopathological Evaluation

The small intestine was stained with haematoxylin and eosin (H&E) to study its morphology. Intestinal tissues were processed and paraffin-embedded before being cut into 3 μm thick sections. The tissue sections were deparaffinised and hydrated prior to H&E staining, after which dehydration and clearing of the sections were performed. A light microscope was then used to view the stained areas, followed by image retrieval employing a laser-scanning confocal microscope. Villi length and width and crypt length were determined using Image J software version 1.53a (Betheseda, MD, USA). Slides were also evaluated by histopathologists in a blinded manner using the histomorphological scores for intestinal inflammation adapted from [38].

### 2.6. DNA Isolation

A commercial tissue section DNA isolation kit (FitAmp General Tissue Section DNA Isolation Kit, catalogue no. P-1003, Epigentek Group Inc., Farmingdale, NY, USA) was used to isolate small intestinal DNA from adult offspring, according to the manufacturer’s instructions. Sample lysis was performed on a dry bath, set at 65 °C for 90 min. The genomic DNA was then captured and purified using spin columns prior to washing out with an elution buffer.

### 2.7. Global Small Intestinal DNA Methylation

The commercial kit MethylFlash Global DNA Methylation ELISA easy kit (catalogue no. P-1030) was used to analyse global small intestinal DNA methylation in adult offspring, in accordance with the manufacturer’s guidelines. This kit measures the 5-methylcytosine (5mC, or methylated DNA), which forms when DNA methyltransferases covalently add methyl groups at the 5-carbon of the cytosine rings. Each group consisted of 5 rats, and the percentage of methylated DNA in total DNA was calculated by using the formula 5mC% = ((sample OD − negative control OD)/(slope × amount of input sample DNA)) × 100.

### 2.8. Statistical Analyses

Two-way analysis of variance (ANOVA) was used to evaluate differences between treatment groups for physiological parameters, whereas global DNA methylation and small intestine morphology were analysed by one-way ANOVA. A *p*-value of 0.05 was considered statistically significant, and was determined by Tukey’s post hoc test. The data were analysed using GraphPad Prism version 8.4 and are presented as means ± SEM.

## 3. Results

### 3.1. Prenatal BPA Exposure and/or Postnatal Trans Fat Diet Affects the Physiological Parameters of Offspring

BPA-TFD started to show significantly higher BW than CTL-ND from PNW9 (389.5 ± 10.0 g vs. 339.0 ± 7.2 g; *p* < 0.05), and this difference persisted until PNW13 (505.0 ± 15.6 g vs. 434.2 ± 13.7 g; *p* < 0.05) (Figure 2a; Table 1). On the other hand, no significant difference in BW was demonstrated among offspring from weanlings (PNW3) to young adults (PNW8). A similar trend was observed in the WC of all offspring, whereby the incline in WC from PNW3 to PNW13 was insignificant (Figure 2b; Table 1), which contradicted the BW data.

Substantial differences in water intake between groups were observed from PNW4 until PNW6, whereby BPA-ND showed higher water intake than CTL-TFD at PNW4 (17.5 ± 0.8 vs. 12.3 ± 0.6 mL; *p* < 0.01) and PNW6 (31.0 ± 1.4 vs. 21.3 ± 2.4 mL; *p* < 0.05), and BPA-ND had higher water intake than BPA-TFD at PNW5 (23.5 ± 1.1 vs. 19.0 ± 0.6 mL; *p* < 0.05) and PNW6 (31.0 ± 1.4 vs. 24.2 ± 1.3 mL; *p* < 0.05) (Figure 2c,d; Table 1). Meanwhile, significant differences in food intake were observed a bit later, at PNW7 and PNW9, whereby BPA-ND displayed higher food intake than BPA-TFD at both weeks (29.8 ± 1.2 vs. 24.3 ± 1.3 g; *p* < 0.05 and 29.5 ± 1.1 vs. 22.8 ± 0.9 g; *p* < 0.01, respectively). Overall, all groups demonstrated increases in water and food intake from PNW3 until PNW8, while some groups showed decreases in those parameters from PNW9 until PNW13 (Figure 2c,d; Table 1). Interestingly, although TFD offspring from control and BPA groups generally demonstrated lower food and water intake than their ND counterparts across the growth period, they were consistently heavier than the ND offspring.

### 3.2. Prenatal BPA Exposure and/or Postnatal Trans Fat Diet Alters Crypt Length of Offspring’s Small Intestine

Some adult offspring from the CTL-ND group showed normal crypt size and cellularity, as well as thin and long finger-like projection of villi (Figure 3a,b). Conversely, a number of BPA-ND offspring exhibited focal broadening and shortening of the villi (Figure 3c,d), with an increase in cellularity (stromal cells, lymphocytes, plasma cells, and eosinophils). The effects of the trans fat diet on the small intestine of offspring exposed and not exposed to BPA in utero were comparable, showing broadening and shortening of the villi, an increase in cellularity, and some superficial mucosal erosion (Figure 4b,d and Figure 5a,b). We also observed elongation of the crypts in control offspring on the trans fat diet (Figure 4a and Figure 5c), which can also be found in the BPA-ND group (Figure 3c and Figure 5c).

Further inspection of the general parameters of villi and crypts in the jejunum and ileum of adult offspring showed that the ileal crypt length of BPA-ND (150.5 ± 5.1 μm; Figure 5c) was significantly longer than those of CTL-ND (96.8 ± 8.9 μm; *p* < 0.001) and CTL-TFD (110.4 ± 5.0 μm; *p* < 0.01), and that the ileal crypt length of BPA-TFD (130.3 ± 9.9 μm; *p* < 0.05) was substantially longer in comparison with CTL-ND (96.8 ± 8.9 μm; Figure 5c). As for the villi length and diameter, no significant difference was observed between BPA-exposed and control offspring on both normal and trans fat diets (Figure 5a–c). Similarly, the histomorphological scores for intestinal inflammation revealed that the severity and extent of inflammatory cell infiltration, as well as villous blunting in BPA-exposed and non-exposed offspring, on both normal and trans fat diets, were minimal and comparable (Figure 6a–c). A low density of inflammatory cells in the small intestinal villus stroma is considered normal in a healthy and normal functioning gut. Nonetheless, it is worth noting that BPA offspring on both normal and trans fat diets exhibited shorter ileum villi length, bigger jejunum villi width, longer jejunum crypt length, and more frequent ileum villus blunting than control offspring on a normal diet (Figure 5a–c and Figure 6c).

### 3.3. Prenatal BPA Exposure and/or Postnatal Trans Fat Diet Changes the Global DNA Methylation of Offspring’s Small Intestine

BPA-exposed offspring on a normal diet showed significantly higher global DNA methylation levels (2.898 ± 0.147%; *p* < 0.05) relative to control offspring on both normal (1.973 ± 0.232%) and trans fat (1.913 ± 0.256%) diets (Figure 7). These results may substantiate the changes observed in the small intestine of offspring in Figure 3 and Figure 4, although there was no statistical difference reported for the general intestinal parameters and histomorphological scores except for the ileal crypt lengths of BPA offspring.

## 4. Discussion

In human adults, obesity is defined as having a body mass index over 29.9 kg/m^2^ [39,40]. A systematic review reporting on the association between BPA and cardiometabolic disorders encompassing Asia, Europe, and North America populations from 2008 to 2014 revealed that the probability of individuals suffering from hypertension, diabetes, and obesity (general/abdominal) increased with the urinary BPA levels [41]. Our physiological parameter data showed no significant difference in body weight among offspring from weaning up to PNW8, suggesting that prenatal BPA exposure may not induce obesity at childhood and adolescence. Nonetheless, a meta-analysis on the relationship between BPA and childhood obesity revealed that children with higher BPA levels have an increased risk of developing childhood obesity [42]. Moreover, the obesogenic impact of BPA in our study was only evident in offspring on the trans fat diet, as compared to control offspring on the normal diet, which can be observed from PNW11–13. This finding is in line with the results obtained from previous studies, in which the body weight of BPA-exposed offspring on a normal diet was similar to control offspring of the same diet at PNW17 [43,44]. Meanwhile, BPA offspring on a high fat diet were significantly heavier than their normal diet counterparts and mimicked the weight of the control offspring on a high fat diet. Importantly, BPA offspring on a normal diet only showed a significant difference in body weight compared to control offspring on a normal diet at PNW28, which is twice the age of our animal model [43]. Perhaps, effects of BPA on the physiological parameters of rats could only manifest later in life. Nonetheless, another study demonstrated a substantial increase in body weight in BPA offspring compared to control offspring as early as PNW13, and this effect persisted until PNW24 [45]. Other than that, there was no significant difference in food intake (normal diet) between BPA and control offspring in our study, consistent with previous findings [43,44,45]. Alfin-Slater [46] suggested that there could be a difference in absorption and/or food efficiency if animals from different diets have different body weights, whereas the amount of food consumed is comparable.

The hypothesis above motivated us to study the small intestinal architecture to identify if there were any morphological changes that could affect the absorption of nutrients. The BPA-induced degenerative changes in the small intestinal villi found in our study, although not significant, resembled the findings of other studies, in which we observed some necrosis of mucosa, superficial mucosal erosion, as well as crypt hyperplasia [47,48]. Furthermore, although jejunum villi length of all groups were not significantly different to each other, which corresponds to the previous studies [49,50], the difference exhibited by BPA offspring on both normal and trans fat diets shall be appreciated. This is because BPA is supposed to suppress the proliferation and differentiation of intestinal epithelial cells, leading to the shortening of the villi [49]. In contrast, a high fat diet supports the proliferation and differentiation of intestinal epithelial cells, leading to lengthening of the villi [51]. Moreover, studies have reported that both obese humans and animals exhibited longer villi length as well as intestinal inflammation, which may reflect the increase in absorption and permeability of the intestine, leading to significant weight gain [52,53]. Perhaps, in BPA offspring on the trans fat diet, the effect of BPA on jejunum villi length was cancelled out by the effect of the trans fat diet. Nonetheless, the pro-inflammatory impact of BPA was not apparent in our study, which contradicted the results of a previous study [50]. Apaydin et al. [48] postulated that alterations in the small intestinal architecture could be due to BPA-induced oxidative damage, as shown in several other BPA studies that worked on hepatic, cardiac, and pancreatic tissues, as well as human erythrocytes [54,55,56]. Importantly, research has demonstrated that intestinal adaptation is influenced by the hyperphagia state in obesity instead of body weight and systemic factors, in addition to the presence of bile acids and enzymes coming from the duodenal papilla or the proximal axis of the intestine [21].

Epigenetic phenomena are particularly vulnerable during the perinatal developmental period, whereby profound changes occur in foetuses and infants [57]. These phenomena could be influenced by nutrients and bioactive food components via direct inhibition of enzymes that catalyse the epigenetic mechanisms or by alterations in the availability of substrates involved in the related enzymatic reactions [58]. Global hypermethylation observed in the small intestine of BPA offspring on normal and trans fat diets may explain why their ileum villi lengths were shorter compared to control offspring on a normal diet. These data agree with the traditional hypothesis of DNA methylation, whereby stable repression of a gene is achieved by the methylation of the CpG islands located at the upstream promoter of transcription start sites which are predominantly nonmethylated [59]. Furthermore, a number of other BPA studies had also reported DNA hypermethylation in fish sperm, rat testis, and mouse tail tissue, affecting the development and function of male fertility and offspring’s phenotype, such as coat colour [60,61,62]. On the other hand, the presence of methyl donors such as choline and folic acid in the trans fat diet may interrupt BPA action on DNA methylation, as shown by previous studies [49,63]. Nonetheless, many gastrointestinal diseases exhibit genomic DNA hypomethylation instead [64]. In addition, oligopeptide transfer and absorption in BPA-exposed offspring were disrupted, as the methylation rate of the CpG site on Pept1 (a nutrient transporter gene) promoter region was substantially reduced compared to control offspring [49]. Therefore, future studies should consider determining the methylation at specific gene promoters, either of the intestinal epithelial stem cells, mature epithelial cells such as enterocytes, or smooth muscle cells. This is to ensure that the global DNA methylation that we observed correlates with the proliferation and differentiation of the intestinal epithelial cells and the smooth muscle cells which are involved in gastrointestinal motility.

The other limitation of this study is that the impacts of developmental BPA and postnatal TFD were not investigated in offspring’s other organs, such as the liver. Consequently, we are not able to determine whether the alterations that we observed in the adult offspring were exclusive to the small intestine only or if these exposures affected other organs as well. Nonetheless, one study exhibited the deleterious effects of BPA and high fat diet on the pancreas and liver of adult male mice [43]. Meanwhile, other studies involving the combination of these exposures only focused on a single organ of interest [27,28,29]. Apart from that, this study suffers from the lack of different timepoints of observations as well as the rather moderate length of the experimental period. Timepoints prior to PNW14 may be of lesser benefit in our case since the physiological and general intestinal parameters did not change much throughout this period. However, dysregulation in the global DNA methylation of the BPA offspring determined at PNW14 may be a good first observation timepoint. This is because other changes observed at this time, such as the intestinal architecture, can project to a certain extent as the experimental duration doubles (i.e., at PNW28), as shown in several other studies [35,43,45]. Importantly, the translational relevance of this research is that our results primarily target those exposed to BPA at occupational levels as well as the neonatal and paediatric intensive care patients, which have demonstrated substantially higher BPA levels than the nonexposed groups [65,66,67,68]. Additionally, as far as we are aware, this is the first study to elucidate the prenatal BPA and postnatal TFD impacts on offspring’s health, particularly obesity, via the epigenetic mechanism of the small intestine. This is crucial, considering that most developing countries, including Malaysia, still do not have any mandatory legislation on trans fatty acid levels [69]. In addition, based on previous findings on BPA exposure outcomes in combination with high fat diet or overfeeding [27,28,29,43], we hypothesise that metabolic disruptions programmed by BPA exposure during pregnancy could modify offspring’s intestinal function response to postnatal TFD.

## 5. Conclusions

The findings suggest that offspring’s physiological parameters and intestinal function may be significantly affected by prenatal BPA exposure. However, a postnatal trans fat diet does not exacerbate prenatal BPA exposure effects on offspring’s small intestine and adulthood obesity. No additive or synergistic effects observed in the combined treatment group (prenatal BPA and postnatal trans fat diet) may suggest that there may be compensatory responses to postnatal trans fat diet in the combined treatment group. Paradoxically, an alteration at the molecular level that was global hypermethylation, as well as a change in the histopathology of the small intestine that ensued from prenatal BPA exposure, shall contribute to obesity development. Nonetheless, we did not observe any substantial differences in the anthropometric measurements of the BPA-ND offspring. Perhaps, obesity would only manifest later in life (middle or late adulthood) in BPA-ND offspring; further investigation is needed to determine this.

## Figures and Tables

**Figure 1 nutrients-14-02382-f001:**
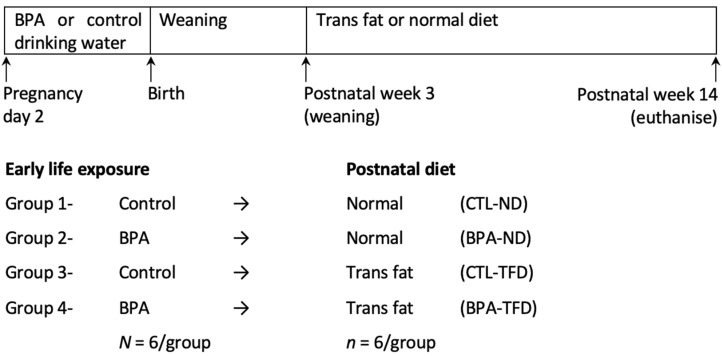
Experimental timeline. Pregnant rats were given either BPA or control (CTL) drinking water throughout pregnancy. Fully weaned offspring were allocated into one out of four study groups (control offspring on normal diet (CTL-ND); BPA-exposed offspring on normal diet (BPA-ND); control offspring on trans fat diet (CTL-TFD); BPA offspring on trans fat diet (BPA-TFD)). Adult offspring were euthanised on postnatal week (PNW) 14 for organ collection.

**Figure 2 nutrients-14-02382-f002:**
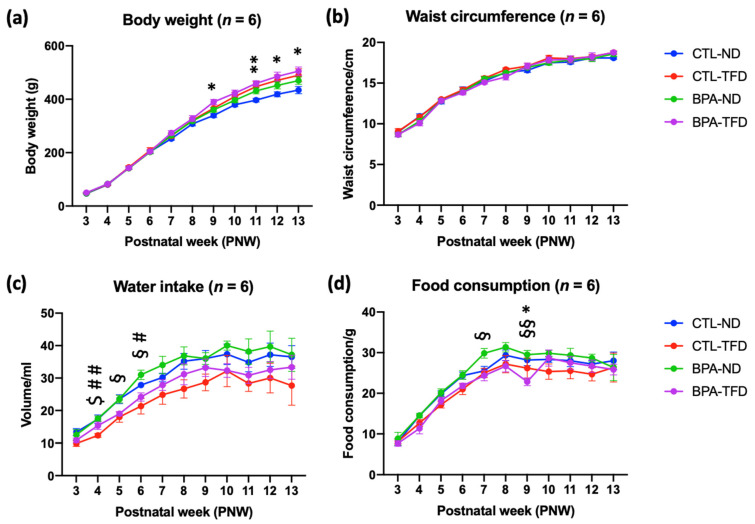
Physiological parameters for obesity in offspring: (**a**) body weight, (**b**) waist circumference, (**c**) water intake, and (**d**) food consumption. Significant differences were observed in water intake first from PNW4–6, followed by food intake from PNW7–9 and body weight from PNW9–13. Interestingly, the waist circumferences of control and BPA offspring on a normal or a trans fat diet were not substantially different from each other. Overall, TFD offspring were heavier than their normal diet counterparts, although they showed lower food and water intake. * *p* < 0.05 CTL-ND vs. BPA-TFD; ** *p* < 0.01 CTL-ND vs. BPA-TFD; $ *p* < 0.05 CTL-ND vs. CTL-TFD; # *p* < 0.05 CTL-TFD vs. BPA-ND; ## *p* < 0.01 CTL-TFD vs. BPA-ND; § *p* < 0.05 BPA-ND vs. BPA-TFD; §§ *p* < 0.01 BPA-ND vs. BPA-TFD (*n* = 6 offspring per group).

**Figure 3 nutrients-14-02382-f003:**
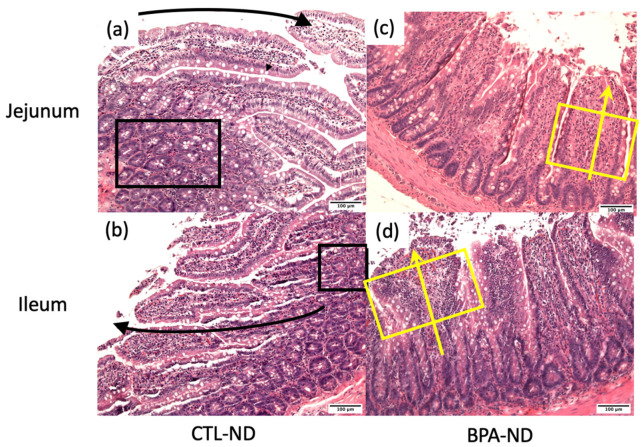
Representative images of the jejunum and ileum sections from adult offspring on a normal diet, stained with H&E (scale bars = 100 μm). (**a**,**b**) Normal jejunum and ileum mucosa in control offspring on a normal diet, respectively. Black curved arrows indicate long and finger-like projection of villi, while black boxes show normal crypt size. (**c**,**d**) Flattening of the jejunum and ileum villi in BPA offspring on a normal diet were focal. Yellow arrows indicate shortened villi, whereas yellow boxes indicate broadened villi. Alterations were not consistent throughout one section and across one group and hence considered non-significant (10× magnification, *n* = 6/group).

**Figure 4 nutrients-14-02382-f004:**
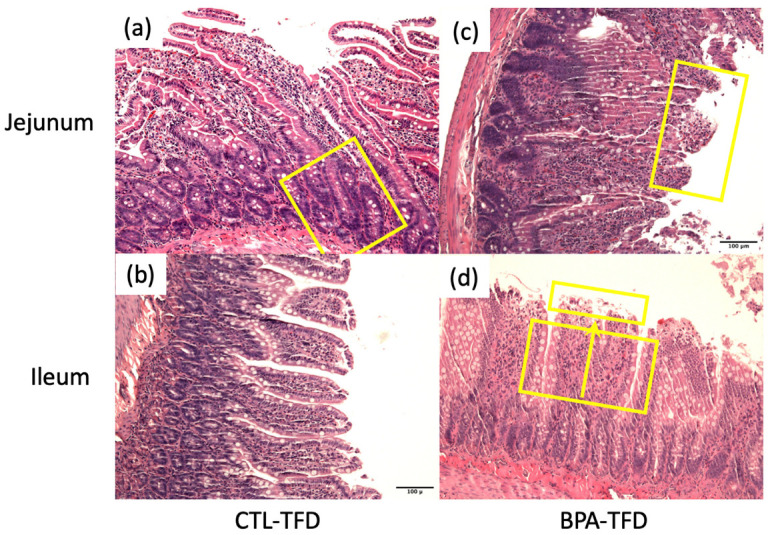
Representative images of the jejunum and ileum sections from adult offspring on the trans fat diet, stained with H&E (scale bars = 100 μm). (**a**,**b**) Elongated crypts (yellow box) and blunted villi were found in control offspring on trans fat diet. (**c**,**d**) Focal broadening and shortening of villi and mucosal necrosis were observed in jejunum and ileum of BPA offspring on a trans fat diet. These changes were scattered across the different tissue sections and therefore regarded as insignificant (10× magnification, *n* = 6/group).

**Figure 5 nutrients-14-02382-f005:**
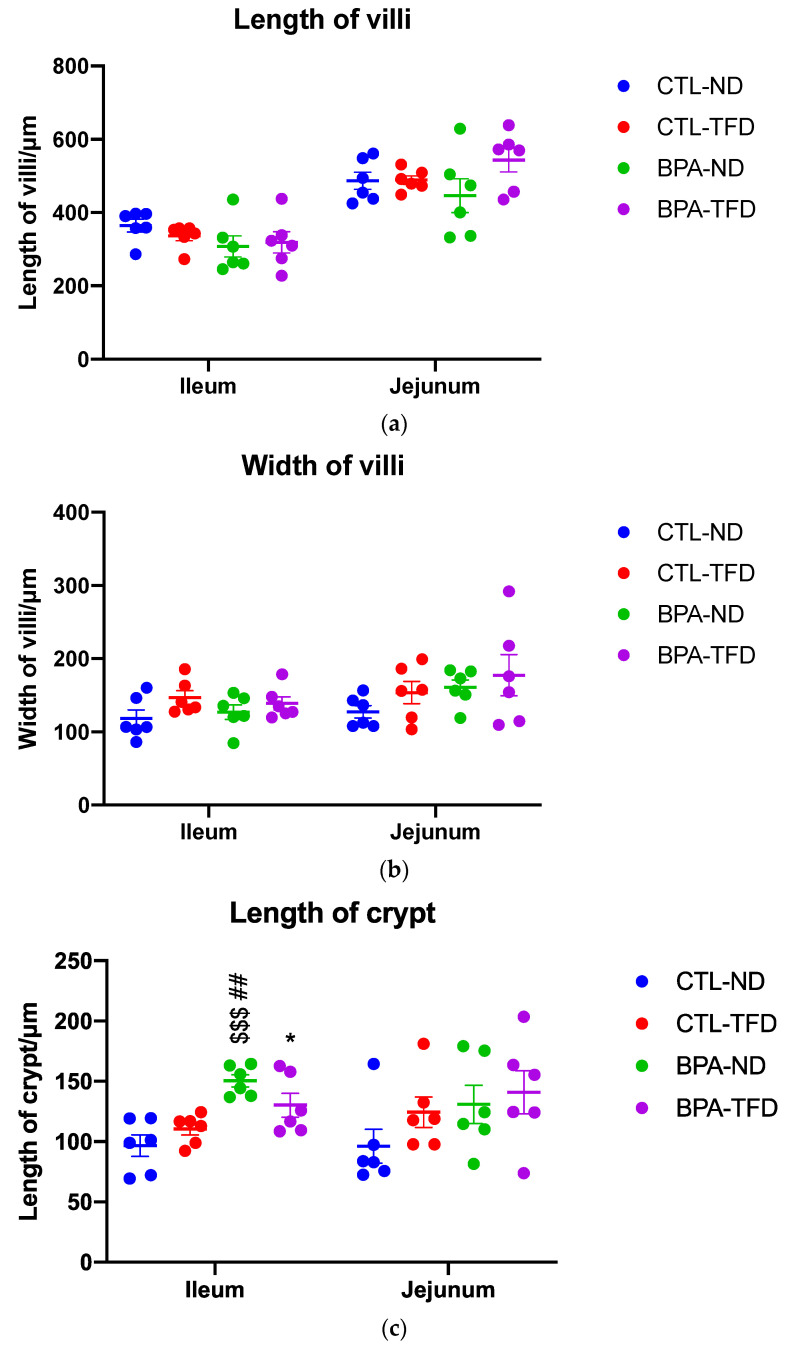
General parameters of villi (length (**a**) and width (**b**)) and crypts (length (**c**)) from jejunum and ileum sections of adult offspring. These parameters were quantitated using Image J software version 1.53a. Each dot indicates individual samples, and data are shown as means ± SEM (* *p* < 0.05 CTL-ND vs. BPA-TFD, ## *p* < 0.01 CTL-TFD vs. BPA-ND, $$$ *p* < 0.001 CTL-ND vs. BPA-ND; *n* = 6/group).

**Figure 6 nutrients-14-02382-f006:**
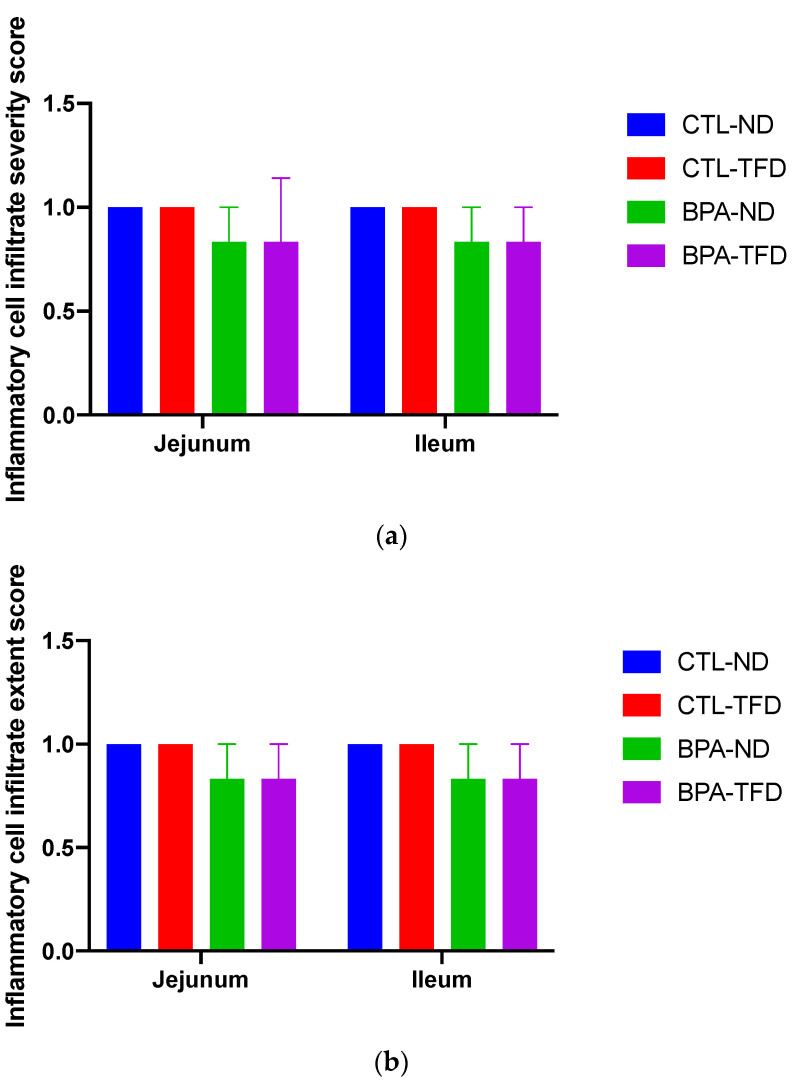
Histomorphological scores of the jejunum and ileum sections from adult offspring. There were three parameters considered during the assessment: (i) leukocyte infiltration, which was divided into the density in the lamina propria area (**a**) and its expansion (**b**), as well as changes in (ii) the epithelium and (iii) the mucosal architecture [38]. Only villous blunting (**c**) was observed under the mucosal architecture category, while other changes such as ulceration, granulation tissue, irregular crypts, and crypt loss were not detected. Similarly, no epithelial changes were observed in all samples except for the jejunum section of one BPA-TFD offspring, which displayed minimal goblet cell loss. The villus blunting score for BPA-TFD offspring’s jejunum was not shown in (**c**) because the value is zero. The score value 1 indicates minimal changes (*n* = 6/group).

**Figure 7 nutrients-14-02382-f007:**
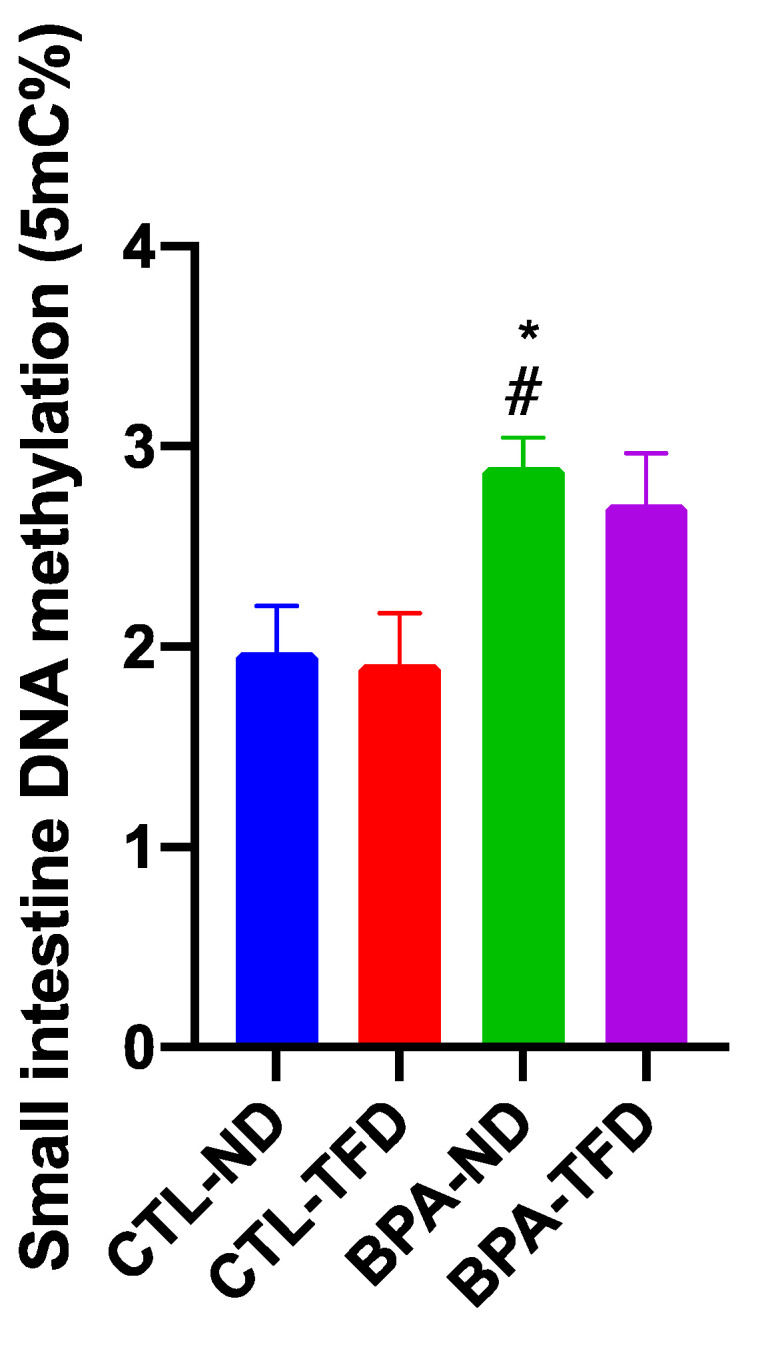
Global DNA methylation in adult offspring’s small intestine (# *p* < 0.05 CTL-TFD vs. BPA-ND, * *p* < 0.05 BPA-ND vs. CTL-ND; *n* = 5 group).

**Table 1 nutrients-14-02382-t001:** Physiological parameters for obesity in offspring (comparisons between groups). Substantial differences in water and food intake were observed at earlier ages (PNW4–9; at adolescence), while significant differences in body weight were only observed at later ages (PNW9–13; at adulthood). Meanwhile, waist circumference was comparable among offspring groups (*n* = 6 per group).

Parameters	Postnatal Week (PNW)	Group			
		CTL-ND	CTL-TFD	BPA-ND	BPA-TFD
Body weight (g)	3	46.8 ± 0.8	48.3 ± 2.0	47.8 ± 2.2	50.0 ± 1.7
	4	80.0 ± 2.2	81.2 ± 4.2	83.0 ± 2.9	82.7 ± 2.9
	5	143.0 ± 4.8	145.2 ± 0.8	141.3 ± 4.0	142.2 ± 3.0
	6	203.5 ± 7.2	208.3 ± 10.4	202.2 ± 5.3	204.2 ± 4.0
	7	252.2 ± 5.6	262.2 ± 7.3	264.7 ± 6.9	273.7 ± 8.4
	8	307.8 ± 7.7	321.3 ± 9.0	318.0 ± 6.2	327.3 ± 11.2
	9	339.0 ± 7.2	366.3 ± 13.4	360.3 ± 8.9	389.5 ± 10.0 *
	10	379.3 ± 7.1	410.2 ± 16.3	396.0 ± 8.1	422.5 ± 11.8
	11	396.7 ± 7.0	447.2 ± 19.8	431.2 ± 11.1	459.2 ± 10.2 **
	12	418.5 ± 9.1	470.7 ± 20.7	451.7 ± 13.0	485.0 ± 16.2 *
	13	434.2 ± 13.7	489.8 ± 24.4	469.8 ± 13.9	505.0 ± 15.6 *
Waist circumference (cm)	3	9.0 ± 0.3	9.1 ± 0.3	8.7 ± 0.2	8.7 ± 0.3
	4	10.9 ± 0.3	10.8 ± 0.3	10.3 ± 0.3	10.1 ± 0.3
	5	12.8 ± 0.3	13.0 ± 0.1	12.8 ± 0.2	12.8 ± 0.2
	6	14.2 ± 0.4	14.2 ± 0.3	13.9 ± 0.2	13.8 ± 0.3
	7	15.3 ± 0.2	15.6 ± 0.3	15.5 ± 0.3	15.1 ± 0.2
	8	16.3 ± 0.3	16.7 ± 0.2	16.3 ± 0.1	15.8 ± 0.4
	9	16.6 ± 0.3	17.1 ± 0.8	16.9 ± 0.2	17.1 ± 0.4
	10	17.5 ± 0.3	18.1 ± 0.3	17.5 ± 0.3	17.8 ± 0.4
	11	17.6 ± 0.2	18.0 ± 0.3	17.8 ± 0.4	17.9 ± 0.4
	12	18.1 ± 0.2	18.3 ± 0.3	18.0 ± 0.4	18.3 ± 0.5
	13	18.1 ± 0.2	18.8 ± 0.2	18.6 ± 0.3	18.8 ± 0.2
Water intake (mL)	3	13.3 ± 1.1	9.8 ± 0.8	12.5 ± 1.6	10.8 ± 0.7
	4	17.3 ± 1.3	12.3 ± 0.6 $	17.5 ± 0.8 ##	15.3 ± 1.1
	5	23.5 ± 1.3	18.0 ± 1.6	23.5 ± 1.1	19.0 ± 0.6 §
	6	27.8 ± 0.9	21.3 ± 2.4	31.0 ± 1.4 #	24.2 ± 1.3 §
	7	30.2 ± 1.3	24.8 ± 2.9	34.0 ± 2.7	27.8 ± 1.6
	8	35.2 ± 2.5	26.7 ± 2.8	36.8 ± 2.8	31.2 ± 3.2
	9	36.0 ± 2.5	28.7 ± 2.6	36.0 ± 1.8	33.2 ± 2.7
	10	37.3 ± 3.1	32.2 ± 4.8	40.0 ± 1.4	32.3 ± 2.1
	11	34.8 ± 2.9	28.3 ± 2.4	38.2 ± 3.9	30.8 ± 2.5
	12	37.2 ± 3.6	30.0 ± 4.6	39.7 ± 4.8	32.5 ± 2.6
	13	36.5 ± 3.5	27.7 ± 6.0	37.2 ± 5.1	33.3 ± 3.7
Food intake (g)	3	8.0 ± 0.8	8.2 ± 0.5	8.8 ± 1.5	7.7 ± 0.7
	4	14.5 ± 0.6	12.7 ± 0.8	14.5 ± 0.6	11.3 ± 1.3
	5	19.8 ± 0.8	17.2 ± 0.8	20.2 ± 0.9	18.2 ± 0.9
	6	24.3 ± 0.9	21.0 ± 1.3	24.5 ± 1.1	21.8 ± 0.7
	7	25.5 ± 1.1	25.0 ± 1.2	29.8 ± 1.2	24.3 ± 1.3 §
	8	29.3 ± 1.3	27.2 ± 2.1	31.3 ± 1.1	26.7 ± 1.4
	9	28.2 ± 1.4	26.2 ± 2.5	29.5 ± 1.1	22.8 ± 0.9 *§§
	10	28.3 ± 0.7	25.3 ± 1.9	29.8 ± 0.7	28.7 ± 2.0
	11	28.0 ± 1.5	25.5 ± 1.9	29.3 ± 1.7	27.5 ± 1.6
	12	27.2 ± 0.8	24.7 ± 1.6	28.7 ± 1.0	26.7 ± 1.5
	13	28.0 ± 2.2	26.3 ± 3.6	26.3 ± 3.2	25.8 ± 1.3

* *p* < 0.05 CTL-ND vs. BPA-TFD; ** *p* < 0.01 CTL-ND vs. BPA-TFD; $ *p* < 0.05 CTL-ND vs. CTL-TFD; # *p* < 0.05 CTL-TFD vs. BPA-ND; ## *p* < 0.01 CTL-TFD vs. BPA-ND; § *p* < 0.05 BPA-ND vs. BPA-TFD; §§ *p* < 0.01 BPA-ND vs. BPA-TFD.

## Data Availability

Not applicable.

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
