# Peer review of "Prenatal Bisphenol a Exposure and Postnatal Trans Fat Diet Alter Small Intestinal Morphology and Its Global DNA Methylation in Male Sprague-Dawley Rats, Leading to Obesity Development"

_nutrients, 2022, doi:10.3390/nu14122382_

Round 1

Reviewer 1 Report

The article is well-written, very interesting and topical. Certainly, it is the starting point for clinical trials. Nevertheless, the results of studies carried out in animals often differ in humans due to a number of differences in the physiology and anatomy of the species, but also in the mechanisms of fetal programming. However, any scientific contribution that aims to reduce the prevalence of obesity in children and adults is important.

Please add your comments and observations (for example in points) that could be used in human research as a summary of your discussion.

Are there any articles that have outlined the the effects of PRENATAL trans fat diet on BPA exposures during pregnancy?

There are some abbreviations in the manuscript, which should be expanded (for example: BPA -line 25).

Author Response

Dear Editor,

We thank the editorial office of Nutrients and the reviewers for their most insightful comments and invaluable input. We have made the suggested changes.  (C stands for comment, R stands for response).  All modifications to our manuscript are in red fonts.

We hope that the revised manuscript meets these expert reviewers’ expectations.

Reviewer 1

C1: Please add your comments and observations (for example in points) that could be used in human research as a summary of your discussion.

R1: Thank you for the comment and suggestion. Translational relevance has been added to the discussion section (page 15, lines 421–424).

C2: Are there any articles that have outlined the effects of PRENATAL trans fat diet on BPA exposures during pregnancy?

R2: Thank you for the comment. As far as we know, no studies have determined the effects of trans fat given prenatally, postnatally or perinatally, in combination with prenatal BPA exposure, in any experimental models. Hence, to the best of our knowledge, we are the first to find out the effects of both exposures on health, via intestinal-related mechanism (page 15, lines 425–429).

C3: There are some abbreviations in the manuscript, which should be expanded (for example: BPA -line 25).

R3: Thank you for the comment. Changes have been made accordingly (page 1, lines 25, 30–32; page 3, lines 111 and 137; page 4, lines 146, 150 and 152).

In closing, we thank the editorial office and reviewer for the helpful recommendations and for giving us the opportunity to resubmit.

Assoc. Prof. Dr Noor Shafina Mohd Nor                               Sarah Zulkifli

Faculty of Medicine,                                                              Institute of Medical Molecular Biotechnology,

University Teknologi MARA (UiTM),                                  University Teknologi MARA (UiTM),

Sungai Buloh, Malaysia.                                                        Sungai Buloh, Malaysia.

Reviewer 2 Report

Prenatal Bisphenol A Exposure and Postnatal Trans Fat Diet Al- 2 ter Small Intestinal Morphology and Its Global DNA Methylation in Male Sprague-Dawley Rats, Leading to Obesity Development

Submitted to MPDI Nutients-May-2022 

Zulkifli et al investigated the effects of postnatal trans-fat diet (TFD) on prenatal BPA exposure effects on offspring’s small intestine and adulthood obesity.  Twelve pregnant rats were administered with either unspiked drinking water (control; CTL) or BPA-spiked drinking water throughout pregnancy. Twelve weaned pups from each pregnancy group were then given either normal diet (ND) or TFD from postnatal week (PNW) 3 till PNW14; divided into CTL-ND, CTL-TFD, BPA-ND, BPA-TFD groups. Body weight (BW), waist circumference, food and water intake were measured weekly in offspring. At PNW14, small intestines were collected for global DNA methylation and histological analyses. They observed marked differences in BW water and food intake differed between offspring at earlier ages only (PNW4–6 and PNW7–9, respectively). Furthermore, morphological changes showed significant elongation of ileum crypt in BPA treated group. Moreover, they show significant global small intestinal hypermethylation in BPA-ND when compared to CTL-ND and CTL-TFD.  The study concludes that Prenatal BPA exposure may significantly affect offspring’s physiological parameters and intestinal function and suggested that there might be compensatory responses to postnatal TFD in the combined BPA prenatal group (BPA-TFD). Of note, postnatal trans-fat diet does not aggravate prenatal BPA exposure effects.

The study is very well-conducted and nicely written.

This study is fascinating and has novelty. The conclusion of this study is very supportive of the data presented here. I have some comments, as mentioned below.

Methods:

1.       Please add references to justify why the authors are using BPA at 5mg/kg/day in the materials and methods. In the introduction (line 89), it is mentioned that this is a high concentration and tolerable concentration is 50μg/kg/day (line 91).

2.       Please add details of Image quantification using image j into the methods section. As the representative images in figures 2 and 3 don’t really match with the quantifications in figure 5. For example, the images show clear difference in villi size and width, but the measured values are not significant. If this was due to animal variations, then the error bars wouldn’t be this tight. You could possibly change the graphs to plotting individual values.

Results:

3.       Figure 5ab and figure 6 finding are not statistically significant which don’t support the statements presented in section 3.2 such is shorter and broader villi, increased cellularity and superficial mucosal erosion.

4.       Signs of statistical significance between the groups are very confusing, please use standard signs such as asterisks or hashtags.

5.       Did the authors investigate the effect of BPA in combination with TFD on any other organs such as the liver to ensure that these effects are specific to small intestine?

6.       The discussion will benefit from a section on limitations of the study, commenting on experimental design and timeline in relation to the observed findings. 

Author Response

Dear Editor,

We thank the editorial office of Nutrients and the reviewers for their most insightful comments and invaluable input. We have made the suggested changes.  (C stands for comment, R stands for response).  All modifications to our manuscript are in red fonts.

We hope that the revised manuscript meets these expert reviewers’ expectations.

Reviewer 2

C1: Methods: Please add references to justify why the authors are using BPA at 5mg/kg/day in the materials and methods. In the introduction (line 89), it is mentioned that this is a high concentration and tolerable concentration is 50μg/kg/day (line 91).

R1: Thank you for the comment. Relevant references have been added accordingly to justify the usage of 5mg/kg/day BPA dosage (page 3, line 130–133).

C2: Methods: Please add details of Image quantification using image j into the methods section. As the representative images in figures 2 and 3 don’t really match with the quantifications in figure 5. For example, the images show clear difference in villi size and width, but the measured values are not significant. If this was due to animal variations, then the error bars wouldn’t be this tight. You could possibly change the graphs to plotting individual values.

R2: Thank you for the comments and suggestion. Image J usage has been added into the methods and results sections (page 4, line 176; page 11, lines 297–298). Graphs for the intestinal parameter quantifications have been changed to plots with individual values (page 11, lines 293–295).

C3: Results: Figure 5ab and figure 6 finding are not statistically significant which don’t support the statements presented in section 3.2 such as shorter and broader villi, increased cellularity and superficial mucosal erosion.

R3: Thank you for the comment. Despite the changes above were not prominent as they were observed in some parts of the sections only, we feel that they shall not be disregarded because there may be a justification point for the modifications of the global DNA methylation levels. To clear out possible confusion from the readers, an explanation has been added to section 3.3 (page 13, lines 321–324).

C4: Results: Signs of statistical significance between the groups are very confusing, please use standard signs such as asterisks or hashtags.

R4: Thank you for the comment and suggestion. “+” signs in Figure 2 and Table 1 have been changed to “§” (page 6, lines 223 and 231; pages 7 and 8, lines 244–245). “%” signs in Figure 5 have been changed to “$” (page 11, lines 295 and 299), while in Figure 7 have been changed to “*” (page 13, lines 325 and 327).

C5: Results: Did the authors investigate the effect of BPA in combination with TFD on any other organs such as the liver to ensure that these effects are specific to small intestine?

R5: Thank you for the comment. Unfortunately, this has not been investigated; therefore, we added this question as part of our study limitations and will become our future reference (page 15, lines 406–412).

C6: Results: The discussion will benefit from a section on limitations of the study, commenting on experimental design and timeline in relation to the observed findings.

R6: Thank you for the comment and suggestion. Limitations corresponding to the experimental design and timeline have been added to the discussion section (page 15, lines 413–420).

In closing, we thank the editorial office and reviewer for the helpful recommendations and for giving us the opportunity to resubmit.

Assoc. Prof. Dr Noor Shafina Mohd Nor                               Sarah Zulkifli

Faculty of Medicine,                                                              Institute of Medical Molecular Biotechnology,

University Teknologi MARA (UiTM),                                  University Teknologi MARA (UiTM),

Sungai Buloh, Malaysia.                                                        Sungai Buloh, Malaysia.